# Strain Improvement and Strain Maintenance Revisited. The Use of *Actinoplanes teichomyceticus* ATCC 31121 Protoplasts in the Identification of Candidates for Enhanced Teicoplanin Production

**DOI:** 10.3390/antibiotics11010024

**Published:** 2021-12-27

**Authors:** Luca Mellere, Adriana Bava, Carmine Capozzoli, Paola Branduardi, Francesca Berini, Fabrizio Beltrametti

**Affiliations:** 1BioC-CheM Solutions S.r.l., Via R. Lepetit 34, 21040 Gerenzano, Italy; lmellere@bioc-chemsolutions.com (L.M.); abava@bioc-chemsolutions.com (A.B.); ccapozzoli@bioc-chemsolutions.com (C.C.); 2Dipartimento di Biotecnologie e Bioscienze, Università degli Studi di Milano-Bicocca, Piazza della Scienza 2, 20126 Milan, Italy; paola.branduardi@unimib.it; 3Department of Biotechnology and Life Sciences, University of Insubria, Via J. H. Dunant 3, 21100 Varese, Italy; f.berini@uninsubria.it

**Keywords:** teicoplanin, division of labor, protoplasts, *Actinoplanes teichomyceticus*, actinomycetes, sterile caste

## Abstract

Multicellular cooperation in actinomycetes is a division of labor-based beneficial trait where phenotypically specialized clonal subpopulations, or genetically distinct lineages, perform complementary tasks. The division of labor improves the access to nutrients and optimizes reproductive and vegetative tasks while reducing the costly production of secondary metabolites and/or of secreted enzymes. In this study, we took advantage of the possibility to isolate genetically distinct lineages deriving from the division of labor, for the isolation of heterogeneous teicoplanin producer phenotypes from *Actinoplanes teichomyceticus* ATCC 31121. In order to efficiently separate phenotypes and associated genomes, we produced and regenerated protoplasts. This approach turned out to be a rapid and effective strain improvement method, as it allowed the identification of those phenotypes in the population that produced higher teicoplanin amounts. Interestingly, a heterogeneous teicoplanin complex productivity pattern was also identified among the clones. This study suggests that strain improvement and strain maintenance should be integrated with the use of protoplasts as a strategy to unravel the hidden industrial potential of vegetative mycelium.

## 1. Introduction

Division of labor is a common feature in natural systems and can be found at different levels of biological organization, from the individuals in a shared society to the cells of a single multicellular organism. In the microbial world, examples of division of labor among colony subpopulations that specialize to perform different cooperative tasks have been extensively described [1,2]. The allocation of tasks can be achieved either at the phenotypic or at the genotypic level [3]. Filamentous actinomycetes are microorganisms that grow to form complex structures in which the alternation of vegetative and reproductive growth phases represents an example of division of labor. Indeed, the vegetative hyphae are programmed to support and protect the reproductive hyphae and the uni-genomic spores. Spores can then persist and disseminate, to allow the spread of the species [4,5,6]. In the model actinomycete *Streptomyces coelicolor*, it was reported that metabolically costly antibiotic production and secretion represents a trade-off with growth, and is performed by only a fraction of the hyphae, which could eventually lose the possibility to produce spores and to propagate, hence identified as the “sterile caste” [7,8]. As the sterile caste displays a maximization in the diversity and production of secreted antibiotics [7], the capacity to identify and preserve it in an industrial setting may represent an opportunity for selecting industrially relevant lineages (i.e., high producers or producers of new antibiotics) [8].

*Actinoplanes teichomyceticus* ATCC 31121 is a “rare” or “uncommon” actinomycete (a group of filamentous actinomycetes other than *Streptomyces* spp., which are quite difficult to isolate, cultivate and genetically manipulate [9]), producer of the clinically relevant lipoglycopeptide antibiotic teicoplanin. Teicoplanin has bactericidal activity against Gram-positive aerobic and anaerobic bacteria including Methicillin Resistant *Staphylococcus aureus* (MRSA) [10], and was recently shown to target the main protease (MPro) in the SARS-CoV-2 virus [11,12]. Teicoplanin was first approved for marketing in Italy as Targocid, with an International Birth Date (IBD) of 30 July 1987. Teicoplanin is a complex antibiotic, consisting of six closely related major subcomponents (TA2-1 to TA2-5 and TA3, where T stands for teicoplanin) (Appendix A) as defined in the European Pharmacopoeia (Ph. Eur.) monograph for teicoplanin [13]. Despite being on the market for more than 30 years, teicoplanin is still a drug of last resort used with success in clinical settings due to its antimicrobial activity and clinical safety. For the above reasons, our laboratories are actively working on teicoplanin production by *A. teichomyceticus* and we are continuously running strain improvement and strain maintenance programs. When dealing with strain maintenance, we reasoned about the potential advantages of identifying and preserving genetically stable high teicoplanin producers within the “sterile caste” lineages. Furthermore, the investigation and identification of a possible lineage-dependent production of the different components of teicoplanin complex could be of particular interest in order to ensure a consistent and reproducible complex composition, in line with the requirements of the Ph. Eur. To achieve these targets, we (i) identified protoplast preparation as a general system for the production of uni-genomic cells (not constrained by the possibility to produce spores), (ii) compared the productivities and complex composition of sub-populations deriving from uni-genomic cells and standard multi-genomic aggregates and (iii) verified the stability of the uni-genomic cell-derived lineages.

## 2. Results

### 2.1. Actinoplanes teichomyceticus Protoplast Production and Regeneration as the Base for Genome Separation

The possibility to separate genomes (and the related phenotypes) within the complex population of *A. teichomyceticus* can be of paramount importance for industrial purposes, to select only those members of the population actively devoted to teicoplanin production. However, the complex structure of hyphae causes the accumulation of genomes in clumps that are difficult to separate. Therefore, the productivity data collected are usually (quantitatively and qualitatively) determined by the average performance induced by the concurrence of different lineages. To efficiently separate (potentially) non-spore forming single genomes from the mycelium of *A. teichomyceticus* ATCC 31121, the production and regeneration of protoplasts was selected as the most promising method.

The filamentous *A. teichomyceticus*, growing in liquid media as tough pellets consisting of aggregating hyphae, presents several cell wall modifications that can result in great resistance to lysozyme [14]. As a consequence of this peculiar growth, when treated with standard procedures for protoplast generation, the cell wall is poorly accessible to enzymatic hydrolysis, and mycelium is scarcely converted into protoplasts, thus resulting in high contamination by hyphal fragments [15]. To improve protoplast production and to reduce hyphal contamination, *A. teichomyceticus* mycelium was treated with a mixture of lysozyme and lipase as described in the Materials and Methods section. The efficiency of protoplast formation was assayed by microscopic enumeration at different intervals of incubation in the digestion solution. Maximum protoplast yield (10^6^/10^7^ protoplasts per ml culture) was achieved after incubation times ranging from 24 to 48 h (Figure 1a). In the model actinomycete *S. coelicolor*, mycelium development has been associated with a progressive movement of replicated chromosomes towards hyphae tips and new branching, according to a mechanism known as nucleoid migration [16,17]. The result is a homogeneous distribution of nucleoids along the whole length of the sporogenic hyphae. The segregation of single genomes within the spores is a process regulated by the ParA-ParB protein complex and follows the homogeneous distribution of nucleoids [18]. When the unfiltered protoplasts of *A. teichomyceticus* (containing also residual mycelium) were treated with the DNA-binding fluorescent dye 4′,6-diamidine-2-phenylindole (DAPI), fluorescent signals also confirmed such organization of the genetic material for *A. teichomyceticus*. Nucleoids appeared well defined and evenly distributed along the vegetative hyphae, as occurs in *S. coelicolor*. In contrast, according to the fluorescence intensity and to the limited dimension of the protoplasts, it was reasonable to attribute a single genome to the regenerating protoplasts (Figure 1b). The absence of fluorescence in some protoplasts could indicate absence of genetic material and therefore the impossibility to revert to mycelium.

### 2.2. Macroscopic and Microscopic Analysis of Hyphae-Derived and Protoplast-Derived Clones

Clones regenerated from protoplasts and clones derived from simple plating of mycelium clumps did not show relevant differences in growth or in other morphological characteristics when replicated on agar plates. Similarly, when inoculated in vegetative and/or production media, hyphae-derived clones showed no variability in the phenotype; presumably, the presence of a consistent number of genomes concurred in all cases in determining an overall similar morphology (Figure 2b). On the other hand, when each independent protoplast-derived clone was inoculated in liquid media, differences in color (Figure 2a), mycelium clump dimension (Figure 2c,d), foaming and overall growth were observed. This result suggested that protoplasts were an efficient tool to separate phenotypes and that the separation was able to uncover an underlying genetically heterogeneous population.

### 2.3. Analysis of Teicoplanin Production and Complex Composition

Teicoplanin is produced by *A. teichomyceticus* as a mixture of related molecules, differing in their alkyl side chain (Appendix A) [19]. For commercializing the pharmaceutical product, two different specifications exist. The first one is described in the European Pharmacopoeia (Ph. Eur.) [13], while the second one is reported in the Japanese Pharmacopoeia (J. P.) [20] (Appendix A). The bottleneck in the industrial production of teicoplanin lies in keeping the balance between the different related molecules (complex of factors) to adhere to the above reported specifications. The characteristics of the complex of factors are strictly dependent, as reported to date, on the composition of the fermentation medium and on the presence of the precursors of the different components of the complex [21]. The fermentation studies performed in this work were aimed at achieving a teicoplanin product in line with the requirements of the Ph. Eur. (an example of an HPLC profile out of the specifications, and one in compliance with the Ph. Eur. are shown in Figure 3a,b, respectively).

In the analysis of teicoplanin production, 49 independent clones derived from hyphal fragments and 49 independent clones derived from protoplast regeneration, were fermented according to the protocol described in the Materials and Methods section. The overall production of teicoplanin (calculated as the sum of the main Ph. Eur. relevant components of the complex, excluding TA3, which derives from downstream processing of the product during industrial purification), and the proportion of the different related complex components were quantified. In fermentations of hyphae-generated clones, the mean teicoplanin productivity was 219.1 mg/L with a standard deviation (SD) of *±* 65.9 mg/L (Appendix A). Teicoplanin productivities ranged from a minimum of 111.0 mg/L (clone H22, Appendix A) to a maximum of 373.7 mg/L (clone H39, Appendix A). Teicoplanin production in protoplast-derived clones ranged from 12.6 mg/L (clone P34, Appendix A) to 508.0 mg/L (clone P73, Appendix A), with a mean of 282.6 mg/L and a SD of *±* 110 mg/L (Appendix A). Hence, compared to what was observed in hyphae-derived clones, the distribution of teicoplanin productivity in protoplast-derived clones was skewed towards both the extremes, with the appearance of both low and high producers (Figure 4, “Complex Sum” panel).

According to the analysis of teicoplanin complex composition (Figure 4 and Appendix A), clones producing a teicoplanin complex compliant with the Ph. Eur. represented 22.4% of the analysed hyphae-derived samples and 12.5% of the analysed protoplast-derived clones. The different factors of the teicoplanin complex were produced in variable amounts with increased variability in protoplast-derived clones (Figure 4 and Appendix A).

The analysis of correlation of the production of the different teicoplanin complex factors evidenced that TA2-1 and TA2-2 abundance in the different hyphae-generated clones were not cross-correlated and were not correlated with the abundance of the TA2-3, TA2-4 and TA2-5 factors (for details see Appendix A). However, TA2-2, being the most abundant component of the teicoplanin complex, correlated with the overall teicoplanin production (r = 0,73; *p* < 0.001). On the other hand, production of the TA2-3, TA2-4 and TA2-5 factors was interconnected, and also correlated with the overall teicoplanin production (Appendix A). A similar outcome was observed in protoplast-generated clones (Appendix A). Notably, teicoplanin minor factors, usually expressed at low level, were uncovered by the separation of protoplasts (for an example see Figure 3a the “uncharacterized” peak).

Finally, four independent colonies obtained from a second protoplasting cycle from each of the original clones (see Appendix A) P73 (i.e., clone with the highest productivity recorded), P75 (i.e., clone with a teicoplanin complex in compliance with the Ph. Eur.) and P41 (randomly chosen clone) (see Appendix A) were independently fermented. Productivities and standard deviations were respectively 511.3 *±* 51.1, 334.9 *±* 43.8, and 357.9 *±* 20.1 mg/L. Data achieved showed good consistency among the four colonies in terms of growth, teicoplanin overall productivity (with SD in the range 5–13%) and teicoplanin complex distribution, which was consistent among the four colonies, as well as between them and the original clone from which they derived. These observations supported the genetic stability of protoplast-derived clones and, hence, the usefulness of the approach herein employed for isolating clones with desired characteristics, namely high production rate and/or compliant complex composition.

## 3. Discussion

In an industrial context, hunting for high producers of pharmaceutically relevant metabolites (or, more generically, of secondary or specialized metabolites) from so-called “rare” actinomycetes such as *A. teichomyceticus* still strongly relies on time-consuming protocols of random mutagenesis and selection (our unpublished data). Other classical approaches that can be useful for improving the production rates of clinically relevant molecules are protoplast fusion and whole genome shuffling [22]; while successfully applied to, for instance, improving tylosin production by *Streptomyces fradiae* [23], these strategies have never been explored—at least to the best of our knowledge—for enhancing teicoplanin productivity. Besides these methods, rational genetic manipulation of producer actinomycetes as *A. teichomyceticus* is possible, but restricted by the still limited availability of ad hoc genetic tools for these bacteria [24,25,26].

In the quest for industrial production candidates, only a little effort has been dedicated to the understanding of the intrinsic phenotypic variability of multinucleate microbes as actinomycetes [27], with the drawback that variability in industrial production still occurs with a high incidence, and the loss of high producers is the inevitable destiny of those companies which do not apply rigorous strain maintenance protocols. Antibiotic production is a costly process involving a direct energy trade-off between production and reproductive capacity [28]. It was recently demonstrated that *S. coelicolor*, when forming colonies on solid substrates, has evolved an elegant mechanism of growth based on a division of labor that limits antibiotic production to a fraction of the colony. The delegation of the production of antibiotics to a “sterile caste” (non-spore producing) of the microbial population reduces the overall costs of biosynthesis, maximizes the magnitude and diversity of the produced antibiotics and increases the reproduction efficiency of the “non-sterile caste” (spore producing) [8]. The process that predisposes to the division of labor is based on differential gene expression [29] and on genomic instability, the latter creating phenotypically heterogeneous subpopulations of cells, mainly by means of large and irreversible deletions or amplifications at the chromosomal termini [8,30,31,32,33].

Although heterogeneity is a beneficial trait in natural biological systems, since it is at the origin of adaptation and evolution, it is generally considered a production pitfall in industry, where the reproducibility of microbial-based processes is of fundamental importance. In this study, we reversed this vision and demonstrated that, for industrial applications, the understanding, identification and preservation of the heterogeneous sterile caste of a microbial population could be of paramount importance for strain improvement and strain maintenance. The study was based on embedding genomes of *A. teichomyceticus* in protoplasts and on the analysis of the derived protoplast-generated clones (for a schematic representation of the approach see Appendix A). Considerable differences in the color of the cultures (ascribable to the expression of silenced metabolic pathways and/or to mutations in metabolic pathways which determine the accumulation of intermediates), foam formation and growth were observed. However, no general rule was identified that could be predictive of the presence of high productivity. By separating the different phenotypes and the underlying genomes that are represented in *A. teichomyceticus* mycelial clumps, and by measuring teicoplanin production and the proportion of the different factors of its complex, we observed a situation that could be plausibly reconducted to a division of labor, similarly to that observed in *S. coelicolor*. Indeed, when compared to standard hyphae-derived clones, protoplast-derived fermentations displayed an extended range of teicoplanin productivity, with both low and high producers (the latter being extremely interesting from an industrial point of view). Furthermore, in protoplast-derived clones, the production of a complex of factors distinct from those produced by clones derived from multinucleated hyphae was observed. Such heterogeneous distribution in the production of the different teicoplanin complex factors, could be interpreted as an adaptative response which may help to maximize the diversity of the secreted antibiotics. Indeed, it was reported that the different teicoplanin factors have variable antimicrobial activities [19], and, therefore, their diversity could help in the struggle against competing bacteria attacking the *A. teichomyceticus* growing colony. Not surprisingly, the highest variability in production was observed in the teicoplanin complex factors TA2-1 and TA2-2, which display the highest antimicrobial activity [19]. Their production appeared instead unrelated to the synthesis of the less active TA2-3, TA2-4 and TA2-5, thus suggesting the possibility of an attack-emergency response specifically based on the most active antimicrobials. These results also suggested that, besides being influenced by fermentation medium composition as previously determined [21,34], teicoplanin complex variability has also a genetic base.

We are aware that our approach has some limitations and that further investigations are required for a better understanding of the genetic bases of the variability observed among the protoplast-derived clones herein produced, for instance, to assess if mutations in biosynthetic and/or in regulatory genes could have been the reasons for the differences observed in teicoplanin production among different clones. Furthermore, we cannot exclude the possibility that more than one genome could have been embedded in some protoplasts, thus generating clones with a mixed genetic background. Neither we can rule out that physiological or genetic changes occurring during protoplast formation and regeneration [33] might have concurred, at least for some clones, in determining the differences observed in teicoplanin productivity and complex profile, or in growth pattern. However, what is clear from our study is that integrating strain improvement and strain maintenance approaches with the use of protoplasts can be a successful strategy for unravelling the hidden industrial potential of *A. teichomyceticus* and related multinucleated “rare” actinomycetes. We could also argue that in the future, micromanipulation and bacterial cell sorting will evolve to be applicable to mycelial microorganisms, allowing a more focused isolation of genomes of interest. Finally, we can speculate that applying recurrent cycles of vegetative mycelium growth, protoplast generation and screening to the most promising clones herein selected might be a key in the future for identifying clone(s) with an even higher titer of teicoplanin production than those presented in this study.

Although highly speculative, the present results might also be a cue for providing a different explanation for classical mutagenesis applied to strain improvement. It is indeed accepted that a chemical or physical mutagenic treatment has the result of killing between 90.0% to 99.9% of the treated microorganisms [22]. This implies that when the mutagenic treatment is applied to hyphal fragments, most of the genomes present therein are lethally mutated, with an outcome very similar to the one that we have depicted in this work, i.e., the segregation of single genome cells. Therefore, it cannot be excluded that, at least in some cases, improved mutants might derive from genome separation, mimicked by genome destruction, rather than from random mutations. This is a fascinating aspect that deserves further investigations.

## 4. Materials and Methods

### 4.1. Strains and Cultural Conditions

*Actinoplanes teichomyceticus* ATCC 31121 [35] was obtained from the ATCC public collection. The strain was maintained as a lyophilised Master Cell Bank (MCB). A Working Cell Bank (WCB) was prepared from the first-generation slant originating from the MCB as already described [36]. Cryo-vials from the WCB were thawed at room temperature and 1 mL of the WCB were used to inoculate 10 mL of Medium V (soluble starch (BD, Franklin Lakes, NJ, USA) 24 g/L, dextrose (Roquette, Lestrem, France) 1 g/L, meat extract (Costantino & C, Favria, Italy) 3 g/L, yeast extract (Costantino & C, Favria, Italy) 5 g/L, Bacto-tryptose (BD, Franklin Lakes, USA) 5 g/L, CaCO_3_ (Gamaco, Imerys, Paris, France) 4 g/L) in 50-mL baffled flasks containing 5–10 glass beads of 3-mm of diameter. 1% *w*/*v* agar (Merck KGaA, Darmstadt, Germany) was added to the growth medium as suggested by Hobbs et al. [37] to obtain a better dispersed growth.

The preparation of hyphal fragments for the generation of hyphae-derived clones was performed as follows. Strains were grown in Medium V to the exponential phase (approximately 72 h) at 28 °C with shaking. Mycelium was then harvested by centrifugation, resuspended in 0.9% (*w*/*v*) NaCl, and fragmented by sonication with Vibracell sonicator 400 W model (AL.BRA S.r.l, Milano, Italy) as previously described [38]. The mycelium was either stored in 1 mL aliquots at −80 °C or immediately processed by plating on Medium V plates solidified with 20 g/L agar (HiMedia, Schenzhen, China).

For protoplast preparation, 10% *v*/*v* of the culture grown for 5 days at 28 °C and 180 rpm was inoculated in 100 mL of Medium VSP (soluble starch 24 g/L, dextrose 1 g/L, meat extract 3 g/L, yeast extract 5 g/L, Bacto-tryptose 5 g/L, CaCO_3_ 4 g/L, sucrose (Merck KGaA, Darmstadt, Germany) 103 g/L, L-proline (Merck KGaA, Darmstadt, Germany) 3.5 g/L) [15], and growth was allowed for a further 48–72 h at the same temperature and in agitation conditions.

### 4.2. Protoplast Preparation

*A. teichomyceticus* protoplasts were prepared by modifying the method described in [15] in order to reduce residual contaminating hyphae and increase protoplast number and regeneration efficiency. In brief, ca. 100 mL of growth culture were centrifuged at 3250× *g.* The mycelium was washed once in P medium [39], then 10 g (fresh weight) were suspended in 50 mL of P medium. For cell wall digestion, hen egg white lysozyme (HEWL) (Merck KGaA, Darmstadt, Germany) and *Candida antarctica* Lipase B (Merck KGaA, Darmstadt, Germany) were added at a final concentration of 10 mg/mL and 0.1 mg/mL, respectively. The non-ionic detergent Pluronic (Merck KGaA, Darmstadt, Germany) was supplemented at the final concentration of 0.1 mg/mL. The digestion solution was then incubated at 28 °C with gentle shaking at 50 rpm, for 24 h. Protoplasts were detached from residual mycelium clumps by thoroughly pipetting up and down, then separated from residual hyphal fragments by filtration through 5-μm durapore membrane filters (Merck Millipore, Burlington, MA, USA). The protoplast suspension was then centrifuged at 16,200× *g*, and finally re-suspended in fresh P medium. The formation of protoplasts was monitored by microscopic observation (Zeiss Axioscope, Carl Zeiss, Jena, Germany) at 400× magnification. The total protoplast number was determined by using a Petroff-Hausser counting chamber.

### 4.3. Regeneration of Mycelium from Protoplasts

Mycelium regeneration from protoplasts was performed using the overlay technique suggested by Shirahama et al. [40], already applied to *A. teichomyceticus* [15]. Plates were seeded by pouring 0.5 mL of protoplast suspension on hypertonic M3 medium, then overlaid with VMS0.1 Medium [15]. To assess residual hyphal contamination of protoplast suspensions, control plates, with V0.1 Medium as the under layer and VM0.1 Medium as the upper layer, were seeded [15]. In these media devoid of sucrose, hyphal cells but not protoplasts were able to grow. Colonies regenerated from protoplasts were then processed for the preparation of WCBs. WCBs originating from three selected clones (P41, P73, and P75) were eventually used to perform a new protoplasting cycle: four colonies obtained from each single clone were selected, and WCB production was carried out as above described for stability testing.

### 4.4. DAPI Staining of Mycelium and Protoplasts

Unfiltered protoplasts (containing also residual mycelium) were treated with 4′,6-diamidine-2-phenylindole (DAPI) dye, solubilized in isotonic P medium at a concentration of 1 mg/mL and incubated at room temperature in dark conditions. After a 5-min incubation, fresh samples were observed by the use of an optical fluorescence microscope Zeiss Axioscope at 254 nm.

### 4.5. Teicoplanin Production and Analysis

Tests for teicoplanin production were performed, starting from WCBs produced from colonies generated from hyphae or protoplasts. 1 mL for each individual WCB was inoculated in 30 mL of vegetative medium (meat extract (Lab M Ltd., Heywood, UK) 4 g/L, meat peptone (Costantino & C, Favria, Italy) 4 g/L, autolysed yeast (HiMedia, Schenzhen, China) 1 g/L, NaCl (Carlo Erba Reagents Srl, Cornaredo, Italy) 2.5 g/L, soybean meal (Cargill Srl, Wayzata, MN, USA) 10 g/L, CaCO_3_ (Baslini Spa, Milan, Italy) 5 g/L, glucose (Roquette, Lestrem, France) 27.5 g/L) in 250 mL DIN ISO 24450 certified unbaffled flasks, and grown for 48 h at 28 °C on a rotary shaker at 240 rpm. 3 mL of the vegetative culture were then transferred to 30 mL of productive medium (glucose 12 g/L, autolysed yeast 4 g/L, malt extract (Costantino & C, Favria, Italy) 35 g/L, cotton meal (Mucedola Srl, Settimo Milanese, Italy) 11 g/L) in 250 mL DIN ISO 24450 certified unbaffled flasks. Flasks were incubated at 28 °C and 240 rpm. At regular time intervals, total teicoplanin was extracted by mixing 1 volume of productive culture broth and 1 volume of acetone. Samples were then centrifuged (16,200× *g* for 10 min) and the teicoplanin-containing supernatant was filtered through a Durapore membrane filter (pore size, 0.45 mm; Merck Millipore, Burlington, MA, USA).

HPLC analyses for quantifying teicoplanin production were performed with the method described in the Ph. Eur. [13], on a 5 μm-particle-size Hypersil ODS (Thermo Fisher Scientific, Waltham, MA, USA) column (4.6 by 250 mm) with elution at a flow rate of 2.3 mL/min with a 30-min linear gradient from 50% *v*/*v* to 90% *v*/*v* phase B. Phase A was 6.89 g/L NaH_2_PO_4_ (Merck KGaA, Darmstadt, Germany) pH 6/acetonitrile (Carlo Erba Reagents Srl, Cornaredo, Italy) 9:1 (*v*/*v*) and phase B was 6.89 g/L NaH_2_PO_4_ (Merck KGaA, Darmstadt, Germany) pH 6/acetonitrile (Carlo Erba Reagents Srl, Cornaredo, Italy) 3:7 (*v*/*v*). Chromatography was performed with a 1100 HPLC system (Hewlett-Packard, Palo Alto, CA, USA), and UV detection at 254 nm. Pure samples of teicoplanin were used as the reference standard (EDQM, Strasbourg, France). Within this paper, we refer to teicoplanin as the sum of the related molecules as defined in the European Pharmacopoeia (Ph. Eur.) document [13], as well as to the single factors or group of factors. For the estimation of compliance of the isolated clones with a potential industrial application, the reference limits for each teicoplanin complex component are those described by the Ph. Eur. [13].

Statistical analysis was performed with the R statistical package [41].

## 5. Conclusions

In this study, we demonstrated that various phenotypes naturally occur in cultures of the teicoplanin producer *A. teichomyceticus* ATCC 31121. A possible explanation of this behavior is the presence of a division of labor among cells, as already observed in the model organism *S. coelicolor*. By producing and regenerating protoplasts, this underlying genetic and phenotypic variability can be uncovered; hence, protoplast preparation might be used as an alternative and convenient tool to select teicoplanin producers with desirable characteristics, i.e., high productivity and/or complex profiles in line with the Ph. Eur. requirements. This approach is valid not only for *A. teichomyceticus,* but is in principle applicable to any filamentous actinomycete of industrial interest for which protoplasts can be produced, including those for which tools and protocols for genetic manipulation are absent or only in their infancy. With our method, we could simplify the strain maintenance work, at least to the extent to which mutations conferring the high productivity traits are sufficiently stable. This paves the way to a new approach to the protocols of strain improvement and strain maintenance.

## Figures and Tables

**Figure 1 antibiotics-11-00024-f001:**
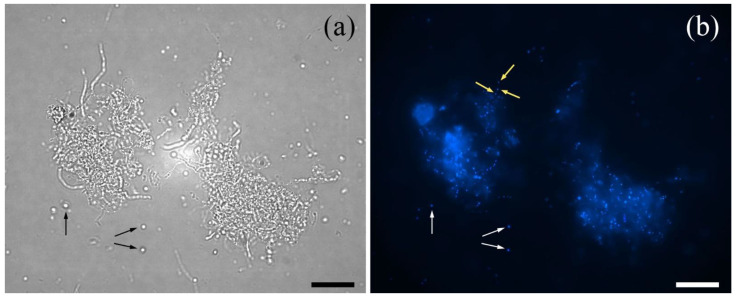
Mycelium clumps and protoplasts (examples of the latter are indicated by black and white arrows in panel (**a**,**b**), respectively), observed with optical fluorescence microscope (Zeiss Axioscope) at 400× magnification (**a**). DAPI staining (**b**) evidenced the complexity of the multinucleated mycelium clumps in comparison to protoplasts and the even distribution of nuclei in the hyphae (as indicated by light yellow arrows). Pictures were taken during protoplast formation (12 h of incubation in the lytic solutions). Bar dimension: 10 μm.

**Figure 2 antibiotics-11-00024-f002:**
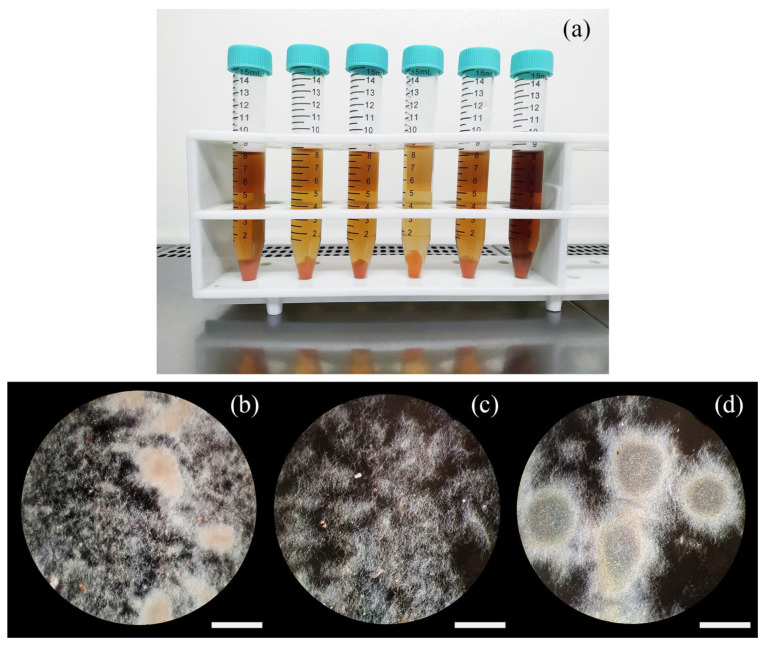
Macroscopic details of cultures derived from protoplast-regenerated clones (**a**). At the microscopic level (**b**–**d**), protoplast-regenerated clones displayed a different degree of mycelium aggregation and clump dimension (exemplified by the extremes in (**c**,**d**)), while hyphae-derived clones were invariably growing in a mixed dispersed-clumped situation (**b**). Bar dimension: 40 μm.

**Figure 3 antibiotics-11-00024-f003:**
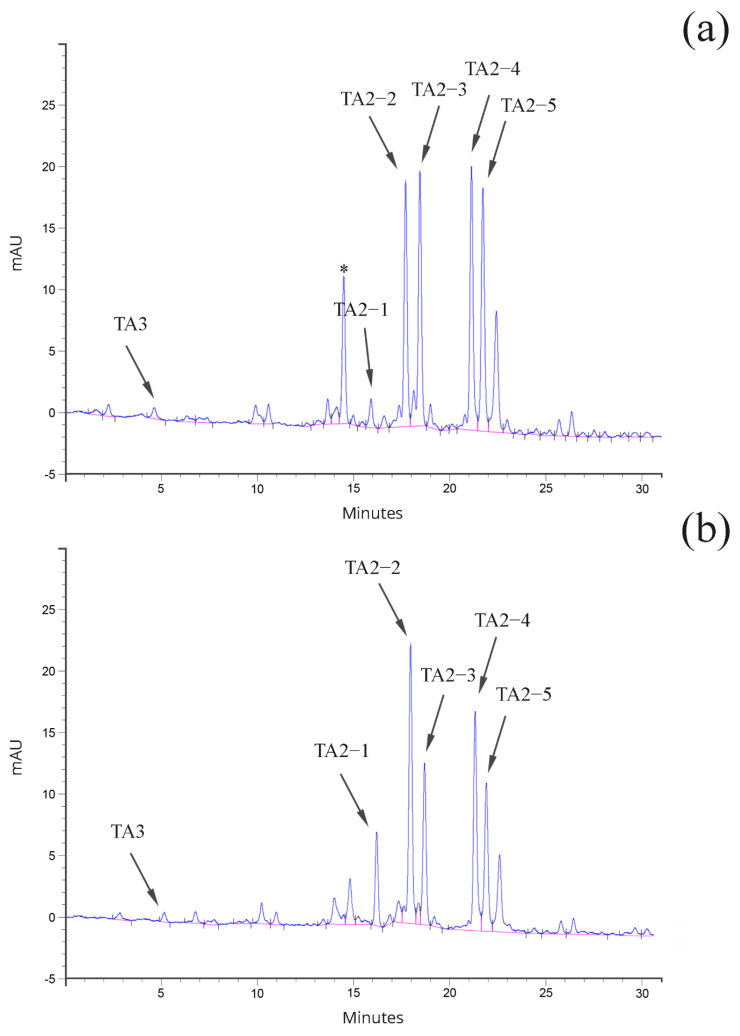
HPLC profile of (**a**) Ph. Eur. out of specification teicoplanin (from the hyphae-derived clone H46), and (**b**) Ph. Eur. compliant teicoplanin (from the protoplast-derived clone P75). Noteworthy is the increase in uncharacterized peaks marked by asterisks in panel (**a**).

**Figure 4 antibiotics-11-00024-f004:**
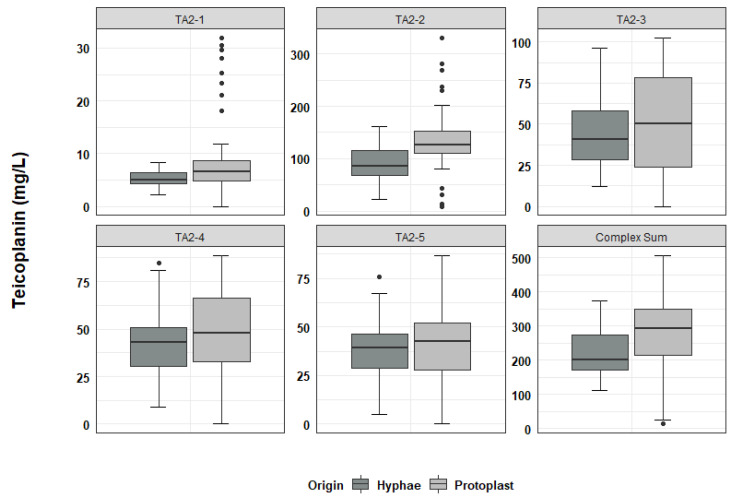
Boxplot distribution of teicoplanin production from fermentation of hyphae-derived clones (Origin: Hyphae; N = 49) and protoplast-derived clones (Origin: Protoplast; N = 49). Dispersion of productivity both in overall production and in single complex factor production was increased in protoplast-derived clones.

## Data Availability

All data are reported in the manuscript or in Appendix A.

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
