# Peer review of "Strain Improvement and Strain Maintenance Revisited. The Use of Actinoplanes teichomyceticus ATCC 31121 Protoplasts in the Identification of Candidates for Enhanced Teicoplanin Production"

_antibiotics, 2021, doi:10.3390/antibiotics11010024_

Round 1
Reviewer 1 Report
The manuscript “Strain improvement and strain maintenance revisited. The use of Actinoplanes teichomyceticus ATCC 31121 protoplasts in the identification of candidates for teicoplanin production” by L. Mellere et al. is devoted to the protoplast regeneration technique for actinomycete strain maintenance. The traditional (by hyphae and spores) and developed (by protoplasts) approaches for bacterial strain maintenance were compared by authors. Regenerates from protoplasts showed higher level of phenotypic variability as we could see from authors’ data. Based on the yields of teicoplanin from only 1 regenerated lineage authors concluded about “the genetic stability of protoplast-derived clones”. Guess, it’s absolutely not enough for this conclusion.
The results obtained by the authors from protoplast-derived clones could be explained by the known fact: “protoplast formation is a stressful process that can lead to profound genetic changes” [Ramijan K. et al. Genome rearrangements and megaplasmid loss in the filamentous bacterium Kitasatospora viridifaciens are associated with protoplast formation and regeneration. Antonie van Leeuwenhoek (2020) 113:825–837 https://doi.org/10.1007/s10482-020-01393-7]. The scientific novelty of the data is questionable but that definitely has wrong interpretation. I think that work should be rejected in the current form.
As well as some minor corrections needed:
line 106: teichomyceticussuch – lost space;
Materials and methods section should be carefully checked: line spacing should be less;
line 409: Ar regular – At;
line 419: NaH2PO4 Merck KGaA, - lost (;
Reviewer 2 Report
The manuscript by Mellere and colleagues reports on the use of protoplast formation and generation as a method to identify variation in teicoplanin production amongst the resultant regenerants. Overall the manuscript is clearly presented and the data support the conclusions drawn. The manuscript reports on the phenomenon (which is a goal for industry) rather than providing any mechanistic insight for the molecular events that underpin the observations.
Scientific Points
- L39, 40. Are these scientific points covered by refs 4-6?
- L50 Please provide a reference for this fact. L271, L281 also
- In my copy, I cannot readily see the well defined evenly distributed signals. Please highlight one in the figure or provide an enhanced inset.
- Do references 29-31 address the differential gene expression? They might be appropriate for the instability. Also, these are all older references. I would imagine that there are more recent studies that, coupled with NGS, would provide greater insight into the changes/instability.
Language revisions/suggestion
- L14 vs L37 vs L42 The authors alternate capitalization styles for actinomycetes
- L22 it allowed the identification of those phenotypes
- Title –perhaps use “..for enhanced teicoplanin production”
- L46 while the other
- L56 SARS
- L68 allows also the investigation and identification
- L90 the cell wall
- L94 the Materials
- L98 associated with a
- L106 confirmed such organization of the genetic material for A. teichomyceticus also.
- L108 In contrast, according
- L141 In contrast or On the other hand
- L176 the mean teicoplanin
- L216 Noteworthy is the increase in the uncharacterized
- L220 represented 22.4% L220 and 12.5%
- L254 correlated either with L257 On the other hand,
- L271 The authors refer to studies that were done “recently” but the reference (22) is from 2001.
- L272 have been applied to
- L316 have variable antimicrobial activities
- L330 Perhaps use lethally mutated? Genomes can not be “killed” per se
- L340 were more than one reference strain obtained? The authors use “collections” rather than one source.
- L344, 346, 347, 359, 360, 361, 362, 363, 371, 372, 373, 396, 402, 406-7, 413, 418, 419, 420. The company locations are only required on first mention.
- L344, 403, 421. Please include the state and use USA rather than US.
- L378 protoplast suspension
- L393 dye, solubilized L394 After a 5-minute
- L396 The microscope details were provided L380.
- L409 At regular L423 Within this paper
- L430 of various phenotypes L431 most probably as a result of a division
- E.P. should be Ph. Eur.
- L438 virtually—not sure if this is the appropriate usage here
- L449, L450 factor production
- The species names (species part) should not be capitalized 466 subtilis, L519, 522, 526, 529, 539 and other instances
- Ref 35 has first names of authors—others do not.
Reviewer 3 Report
The MS present a nice suggestion the use of protoplast to have improved strains of Actinoplanes teichomyceticus. Anyway, the conclusions are not adequately supported by a good presentation of the data.
For example, the last three lines of the abstract are exaggerated; the authors should simply write that using protoplasts can provide even better results than using mycelium or spores.
Moreover:
Line 20: it needs to be modified because phenotypes are separated first and thereafter phenotypic differences may result in distinguished genomes.
Line 96: the yield in protoplasts appears extremely high and in any case it is not understandable why it is indicated "from 100 ml of culture" together with Figure 1. This confuses the reader because the final yield indicated is from 100 ml of culture but Figure 1 certainly represents an intermediate stage of the procedure since mycelium and protoplasts are present together. Therefore it should be indicated which sample / stage of the procedure is represented in Figure 1. Moreover the classical unit is no. / ml.
Figures 3 and 4: Apparently the order is A protoplast - B hyphae in Fig. 3 and Left hyphae - Right protoplast in Fig.4. The same order should always be maintained in order not to confuse the reader. In addition, for Figure 4 it should be confirmed in the legend that the analysis concerns 49 samples.
Figure 3: The peak with the asterisk is a problem because it is so large that it appears to be more than 5% of the components of the Teicoplanin complex; in other words, the hyphae sampled give rise to a product that would not comply with the Ph. Eur. limits, with the consequence that the starting hyphae would not be suitable for the production of Teicoplanin. The conclusion is that it is necessary to characterise the asterisked peak.
Tables S1 and S2: It is appropriate to omit the division into groups, otherwise it would be necessary to explain to the reader what the logic of the division into groups is (an additional figure would be needed) and this is not necessary for the purposes of this paper. Therefore, it is sufficient to limit the information to TA3, TA2-1 .... TA2-5, as done for Figure 3.
Table S1: Please explain the "Relative retention time" as in Figure 3 retention times are expressed from 0 to 30 minutes.
Table S2: Total TA2 must be reported as not less than (NLT).
Figure S1b: for sample p34, 100% TA2-2 is reported. this value raises some doubts, because in nature 100% of a single substance is very, very rare, and vice versa could be an important result. Please check and eventually discuss the data.
Reviewer 4 Report
In the manuscript, Mellere and colleagues initially present an interesting topic that division of labor exists in multicellular bacteria actinomycetes as in eukaryotes. Protoplasts become cellular parts that can be separated to isolate a single genome encoding high levels of teicoplanin. A previously developed method is used to prepare protoplast of a teicoplanin producer (Actinoplane teichomyceticus ATCC31121). Phenotypes and chemotypes are analysed and compared between independent isolates from hyphal fragments and protoplast regeneration. While cultures generated from protoplast produced a wider range of production titer, individual clones with the highest titer could be isolated. Representative colonies from clone P73 are claimed to be genetically stabile protoplast-derived clones. At the time of review, protoplast screening sounds like a new approach to strain prioritization.
Broad comments:
- Representative colonies from clone P73 are claimed to demonstrate genetic stability of protoplast-derived clones. Please include data relevant to the four colonies.
- I agree that high producers are desirable for the industry. It might be useful to provide additional information on how to choose desirable clones in this study (e.g. P75 and P73 in lines 215 and 263 respectively). If Ph. Eur. specification is one of the criteria, I perhaps suggest including the percentage of each teicoplanin factor (relative to the complex) in Table S3 or elsewhere in the supplementary information, so that we could easily follow the production is compliant to Ph. Eur.
- I am not familiar with teicoplanin factors (TA2-1 to TA2-5). Would it be possible to show chemical structures or at least chemical formula?
- Variable readings on teicoplanin production could be resulted from multi factors during fermentation and physiological-genetic changes during protoplast regeneration. I am not fully convinced with some statements concluded by the authors that variations of phenotypes are result of division of labor. I essentially did not see any evidence provided in the manuscript. If it is true for division of labor, would a protoplast-derived clone not go through another division of labor during growth (i.e. some cells are less productive, other cells are more productive)
- Compared to hyphae-derived clones, protoplast-derived clones could achieve higher titer up to ~500 mg/L maximum with average ~280 mg/L. In published literature, alternative methods (e.g. random mutagenesis and genetic manipulation) could possibly obtain stronger titer improvement. I would strongly recommend to explicitly describe why these alternative approaches are less or not feasible to A. teichomyceticus. Additionally, briefly discuss how to further improve production titer using the protoplast isolation method.
Specific comments:
In general, another careful read-through and editing is encouraged for this manuscript.
Figure 4: n=49?
Lines 47-50: a long sentence – try to split the ideas
Lines 71-72: please be specific - teicoplanin production and analogue composition
Line 73: How to verify the stability? Please provide data and methods
Lines 169-170: Were there any replicates being analyzed for each of these 49 independent clones? Would it be better to choose 30 independent clones with two replicates (2 x 30)?
Lines 320-333: confusing as to the connection with this study. Rework is needed
Lines 434: what are E. P. requirements?
Lines 440-442: Were mutations determined in this study?
Round 2
Reviewer 1 Report
In the reply, authors wrote:
We also need to keep the focus on the objective of our paper, which was to understand if the separation of genomes embedded in vegetative mycelium was a convenient strategy for the identification of mutants with interesting industrial characteristics.
- separation of genomes - Authors didn't use micromanipulation or cell sorting techniques and we couldn't see any proofs that regenerates are derived from different types of actinobacterial cells.
- mutants - Authors didn't use sequencing for regenerates and we couldn't see any proofs that regenerates have mutations.
In general, the fact that we can obtain more productive lines using protoplast regeneration approach is very interesting and useful for industrial microbiology. But the authors' interpretation of the obtained data is not adequate and not supported by the data. In the second time I recommend to reject the manuscript in the current form.
Author Response
Answers to reviewer 1
Dear reviewer,
your comments are much appreciated as they are a way to keep improving our manuscript.
Concerning specifically points 1 and 2
- we have attempted sorting the protoplasts with FACS but this was not feasible due to their small size. Actually, filamentous actinomycetes are difficult to sort and their low dimension and complexity make micromanipulation, at present, feasible with aggregates of mycelium or entire sporangi only. We have added a comment regarding this issue in the discussion (lines 358-360). In our study, the separation of protoplasts from mycelium by filtration was used and the result was verified microscopically. The selection of the different phenotypes was simply based on the separation of colony forming units by dilution and plating, which is a trivial but still universally known as trustable technique. Note that we did not claim to have obtained “different types of actinobacterial cells” in any part of our manuscript. Simply we claim to have separated different phenotypes (and the underlying genomes) able to produce different levels and/or complex of teicoplanin. Rather, the issue could be on having more than one genome embedded in a single protoplast, as this could bring us to a mixed genetic background in which the dominant phenotype could prevail. That was a foreseen risk which was taken and we agree that some of our clones could be not absolutely pure as of their genetic background. This still does not prevail on our conclusions.
- As far as it is not due to changes in the fermentation procedure for the identification of the different phenotypes (and the procedure was kept constant in our work), it is hard to assume that there is no mutation responsible for the differences we observed. Differences could be in biosynthesis involved genes or, more probably, in regulators. In any case they are the result of some mutation independently from the fact that no sequencing was performed.
Reviewer 3 Report
The Ms has been corrected and improved but a couple of problems still remain:
a) lines 249-255: it is mentioned the fermentation of three (not 4 as written in the text) colonies obtained from a second cycle of protoplast selection, clones P41, P73, P75 with values of the total Teicoplanin production different from those indicated in Table S7, consequently it would be necessary and useful to present for the three fermentations the HPLC profiles indicating the possible presence of the not characterized peak. In addition, it is necessary to expressly indicate that of the three clones apparently only P75 produces Teicoplanins within the limits of Ph. Eur (Table S7).
b) lines 256-260: 4 colonies are indicated instead of the three for which productivity data were previously reported, while the 'good consistency among the four colonies' and the data supporting 'the genetic stability of protoplast-derived clones' need to be explained.
Author Response
Answers to reviewer 3
We thank very much the reviewer for carefully addressing the point. We have to clarify that the analysis was performed on 4 replicates (see the section materials and methods at lines 396-400 for the details on how the replicates were obtained) for each single selected clone (P41, P73, and P75). As those are replicates, the average value obtained could slightly vary but, the original value obtained after the first cycle of protoplast selection and reported in Table S7, lies within the +/- SD of the fermentations conducted on the four colonies obtained after the second cycle of protoplastization. We have clarified in the text, at lines 245-247, that the 4 replicates were from the same original clone. Please let us know if it is now clear. Note that clones P75 and P41 productivities were accidentally inverted. The mistake has been corrected in the text.
Concerning clone P75, this is the only one, among the three that went through additional studies (P41, P73, and P75), which had a teicoplanin complex profile in line with the requirements of the Ph. Eur. (as specified in lines 247-248). The four replicates of clone P75 still produced a teicoplanin complex according to the Ph.Eur.
Among all the clones tested after the first round of protoplast selection, lines 200-205, indicate the proportion of them which produced a teicoplanin complex in line with Ph.Eur.
Concerning your comment b), the consistency among the 4 colonies still indicates the 4 colonies tested for each of the clones P41, P73 and P75. Consistency is considered good as the SD for the 4 fermented colonies deriving from the same clone displayed a SD below 15%. In order to enhance clarity, also the addressed part was modified. Please see text for details at lines 251-257.
Reviewer 4 Report
The authors have revised the manuscript substantially, improving the clarity of result and discussion sections – Thank you very much for the effort! Now the manuscript is in a better shape for publication. However, listed below are several points for consideration in further edits.
- As stated in author response, protoplast-derived clones can be readily applied in many labs without the need of ‘fancy’ genetic tools. Please further discuss whether teicoplanin production titer could be improved by this approach to the next levels (protoplast screening). For example, would another screening pf protoplasts derived from P73 likely discover a new strain with higher titer of teicoplanin production
- Lines 47-51: I might get the message, but the sentence(s) could be simplified
- Lines 72-74: The sentence is still odd. How the complexity…also allows the investigation…?
- Line 102: Do you mean 106-107 protoplasts per mL culture?
- Lines 108-112: a long sentence – try to split the ideas
- Line 144: add “(data not shown)” after no variability in phenotype. Or maybe, low phenotypic variability instead of no variability? (no variability is too ideal in the lab)
Author Response
Answers to reviewer 4
We thank the reviewer for appreciating our efforts. Below are the answers to each single suggestion.
- As stated in author response, protoplast-derived clones can be readily applied in many labs without the need of ‘fancy’ genetic tools. Please further discuss whether teicoplanin production titer could be improved by this approach to the next levels (protoplast screening). For example, would another screening pf protoplasts derived from P73 likely discover a new strain with higher titer of teicoplanin production
According to previous studies cited in our work, there are several genetic rearrangements which generate the variability in productivity. At present, we can say that the four colonies generated from clone P73 that we randomly selected for further studies after a second round of protoplast formation, showed a productivity level that was comparable to the one of the original P73 clone. However, we could speculate that recurrent growth of vegetative mycelium and protoplasting could actually originate other improvements but this should be proved experimentally. In our long-lasting strain improvement and fermentation development activities we often faced with apparently mysterious outliers in sets of parallel, identical fermentations. The random appearance of mutations and genetic rearrangements (which eventually represent the dominant population in single flasks of parallel identical fermentations) could be an explanation for this effect and was encountered even with strains which already underwent several rounds of chemical mutation and selection. In conclusion, we do not exclude the possibility of further improving productivity through protoplast, but other studies should be performed. A comment relative to this aspect has been introduced in the Discussion at lines 330-334.
- Lines 47-51: I might get the message, but the sentence(s) could be simplified
We have split the sentence in order to simplify the concept
- Lines 72-74: The sentence is still odd. How the complexity…also allows the investigation…?
The sentence has been rephrased according to the reviewer’ suggestion
- Line 102: Do you mean 106-107 protoplasts per mL culture?
Yes, we have modified as indicated
- Lines 108-112: a long sentence – try to split the ideas
We have split and simplified as suggested